# Dynamic-Bayesian-Network-Based Project Cost Overrun Prediction Model

**Sou-Sen Leu \*, Cheng-Yu Lu and Pei-Lin Wu**

Department of Construction Engineering, National Taiwan University of Science and Technology, 43 Keelung Rd., Sec. 4, Taipei 106, Taiwan
\* Correspondence: leuss@mail.ntust.edu.tw

**Abstract:** One common problem in the construction industry is project cost overrun. Cost overrun can have significant impacts on financial profitability, project completion, project quality, and stakeholder satisfaction. The average percentage of construction project overrun can vary widely depending on the project type, size, complexity, and location. Many approaches can be adopted to prevent or mitigate project cost overrun, and one of them is a more accurate cost estimate and prediction. Several studies on the construction project cost overrun estimation and prediction have been conducted based on historical data; nevertheless, each project has its project characteristics and cost trend. Real-time, project-specific cost data are more reliable for forecasting the cost trend of the project itself. There are many influence factors that may interdependently affect the construction project cost overrun. This paper proposes a real-time predict cost overrun risk prediction model based on the influence factors and their interdependence as well as the corrective actions if adopted. This study used a dynamic Bayesian network (DBN) to formulate problem architecture and to use the input–output hidden Markov method (I/O HMM) with particle filter (PF) to run inference. Six building and mass rapid transit projects in Taiwan were used as model validation and comparison. The posterior probabilities from the DBN-based cost overrun risk prediction model were highly consistent with the cost overrun ratios of real construction projects. Moreover, it is superior to other prediction models in terms of accuracy. The proposed model could provide project managers with an early alert for cost overrun.

**Keywords:** project cost overrun; dynamic Bayesian network; input–output hidden Markov method; particle filter; Bayesian network; sequential simulation

## 1. Introduction

Construction project cost overrun always potentially exists in a construction project's execution. Cost overruns can have significant impacts on financial profitability, project completion, project quality, and stakeholder satisfaction. A cost overrun can result in serious financial losses and profitability for a construction company. Cost overrun could also lead to delays in project completion because the construction company cannot support sufficient construction resources due to a lack of money. Moreover, some construction companies may cut corners or compromise on quality under the situation of cost overrun. Cost overrun could damage the construction company's reputation and lead to a loss of trust from stakeholders, including investors, customers, and employees. The average percentage of construction project overrun can vary widely depending on the project type, size, complexity, and location. Many approaches can be adopted to prevent or mitigate project cost overruns, such as a suitable project plan, in-time project monitoring, and accurate cost estimate and prediction. To more reliably conduct cost estimates and predictions, many factors and their interdependence that influence the construction project cost increase need to be considered. Examples include design ability, procurement approach, and management quality. Each construction project has different influence factors and impact patterns on cost overrun. As a result, the cost overrun prediction needs to consider

the variability of factors and their influence patterns on project costs. Many studies in the construction project cost domain have been conducted to develop methodologies that take the effects of uncertainty on project cost overruns into consideration, such as regression, simulation and artificial neural networks (ANNs) [1–7]. Most of them heavily count on historical data. As discussed above, each project has its own project characteristics and cost trend. It is better to only use historical data as prior information at the beginning of the prediction. During the project operation, real-time project cost data are collected to more accurately forecast its own cost trend. Such prediction by a reliable cost management system can provide trustful warning of cost overruns as early as possible.

As briefly discussed above, past studies have explored potential influence factors of cost overrun; nevertheless, they rarely considered the interdependence among the influence factors on cost overrun. The model presented in this article first used a Bayesian network (BN) to construct a basic model and was then based on I/O HMM to develop a time-sequential DBN architecture. Finally, the PF algorithm was adopted to do the sampling based upon the assessment data. The model attempts to predict cost overruns probability according to the factors that interdependently affect cost overrun. Moreover, in practice, the project manager needs to determine how effective the corrective action is to diminish the cost overrun risk once it is adopted. It is better for a cost overrun prediction model to take the effect of corrective action into consideration.

In this model, only the project's own cost data and status reports are used. The model presented in this research attempts to forecast cost overruns probability based on the interdependent influence factors of project cost overrun and the corrective action if adopted. The significance of the model built is summarized: (1) the model mainly relies on the project cost records to calculate real-time predictions without historical data as prior information; (2) it considers the interdependence among the influence factors; and (3) it can assess the effect of the corrective action after the occurrence of the cost overrun.

This paper is organized as follows. First, a literature review of cost overrun factors affecting cost overrun and their dependence as well as recent cost overrun prediction methods are reviewed. Second, the DBN-based cost overrun model is first explored and established, and then I/O HMM with PF for cost overrun probability inference is presented. Finally, six real building and MRT construction projects with 53 monthly cost data in total were used for model validation and sensitivity analysis. Comparison with other approaches was further conducted to verify the appropriateness of the DBN-based algorithm and to demonstrate the application of the method.

## 2. Literature Review

In previous studies, various statistical and artificial intelligence methods and tools have been used to solve the problem of predicting construction costs and cost overruns in the construction projects, such as regressions, neural networks, machine learning, fuzzy logic, Bayesian networks, simulation, etc. [1–9]. Some more recent research is briefly explained in the following. Ahiaga-Dagbui and Smith [10] used nonparametric bootstrapping and ensemble modeling with 1600 completed projects to develop the project's cost forecasting models for early budget estimates. El-Kholy [4] conducted an expert questionnaire survey containing 44 questions in Egypt, and finally, 11 significant factors were filtered out for the development of regression and CBR models. Huo et al. [5], based on 57 projects, completed in Hong Kong (1985–2015) to identify the significant factors of project cost overrun using a statistical approach. Three major factors were project type, project size, and project duration. Plebankiewicz and Wieczorek [6] adopted fuzzy inference to develop the construction project cost overrun risk prediction model based on three influence factors: the share of element costs in the building costs, predicted changes in the number of works, and expected changes in the unit price. Ashtari et al. [7] conducted an expert questionnaire survey containing 43 questions in Iran, and finally, 10 significant factors were identified for developing the Bayesian network classifier. Based upon the survey mentioned above, the limitations in previous research are described as follows: (1) the previous research

generally developed macro-level models, and they need a lot of historical construction project data or questionnaire data; (2) few past research support the assessment of cost overrun based on real-time, project-owned data (micro-level), including cost statuses and the interdependence among the influence factors; and (3) the previous models seldom consider the effect of corrective action after the occurrence of cost overrun events. Each project has its project characteristics and cost trend; real-time project cost data are more reliable for forecasting its cost trend. The model presented in this paper attempted to develop a prediction model to estimate cost overrun probability founded on consecutive cost statutes, the influence factors with their dependence, and the effect of corrective action if adopted during the project execution.

In addition, several past studies have also been conducted that surveyed the influence factors of project cost overrun. The classification of cost overrun factors is diversified based on the research focuses and purposes [3,9,11–14]. For the overall assessment, macro-level factors are generally defined for the model construction. They can be project scope, project size, project duration, etc. As discussed above, adopting project-owned cost performance data to estimate and control project cost overrun during project execution may be more reliable. These factors belong to the project-specific level (micro-level); i.e., they are generally stepwise assessed and recorded in the project cost reports based on cost performance outcome and the corresponding influence factors. Part of the recent research is summarized in Table 1 and briefly explained in the following. Wang and Demsetz [15] summarized five significant cost overrun factors: approval delay, weather, material delivery, labor, and equipment. In the study of Elhag et al. [12], several external and internal cost overrun factors were summarized, such as client characteristics, consultant and design parameters, contractor attributes, project characteristics, contract procedures, and procurement methods, as well as external factors and market conditions. Aljohani et al. [11] intensively surveyed the causes of construction project overrun based on a literature review. There were 173 causes of cost overrun summarized in seventeen internal and external frameworks. Idrees and Shafiq [14] surveyed the public construction projects in Pakistan using a questionnaire survey and descriptive statistics to explore the significant factors for cost overrun. They were legal issues, technical error, and project management. Xie et al. [3] surveyed critical influence factors in construction projects using fuzzy synthetic evaluation. There were 65 critical factors covered in the research classified into four categories: project macro, project management, project environment, and core stakeholder.

**Table 1.** Summary of project cost overrun influence factors.

| Literature | Influence Factor of Project Cost Overrun | | | | | | | | |
|---|---|---|---|---|---|---|---|---|---|
| | Design Ability | Change Order | Management Quality | Subcontractor Coordination | Project Condition | Procurement Approach | Market Impact | Contract Argument | Force Majeure |
| Yeo (1990) [16] | ◎ | ◎ | ◎ | | | ◎ | ◎ | | ◎ |
| Elinwa and Buba (1993) [17] | | | | | | ◎ | ◎ | ◎ | |
| Kaming et al. (1997) [15] | ◎ | | | | ◎ | ◎ | ◎ | | |
| Akinci and Fischer (1998) [18] | | | ◎ | | ◎ | | | | ◎ |
| Dissanayaka and Kumaraswamy (1999) [19] | ◎ | ◎ | ◎ | ◎ | ◎ | | | ◎ | |
| Wang and Demsetz (2000) [7] | ◎ | | ◎ | ◎ | ◎ | | ◎ | ◎ | |
| Elhag et al. (2005) [20] | ◎ | ◎ | ◎ | | ◎ | ◎ | | ◎ | ◎ |
| Nassar et al. (2005) [21] | ◎ | ◎ | | | | | | ◎ | |

Note: ◎ = factor that affecting cost overrun.

The cost overrun factors were apparently different in the mentioned studies. This research attempts to forecast cost overruns probability based on the project-owned cost performance data, influence factors, and corrective action if adopted. By unifying the factors proposed in the previous studies, these attributes were reclassified based on their common characteristics. Nine significant project-specific classification factors were defined.

They are design ability, change order, management quality, subcontractor coordination, project condition, procurement approach, market impact, contract argument, and force majeure. The project cost tends to overrun if the poor status of these factors occurs during construction project execution. The real-time status of these factors can be summarized and surveyed following the project reports and checklists. Based upon these status data as inputs to the model in the following, the cost overrun risk can be in-time assessed, and the effect of the corrective action is also surveyed. According to the survey of cost overrun risk and influence factors, the project management division can timely establish proper effective cost–risk treatment plans to mitigate the chances of cost overrun during construction project operation.

## 3. Materials and Methods

To achieve the abovementioned objectives, DBN was proposed to forecast the cost overrun possibility in accordance to the interdependent factors of cost overrun and the corrective action. I/O HMM was further used to simulate the cost overrun occurrences, on which the influence factors are dependent. In the proposed model the corrective action is also considered as output in the model once the cost overrun occurs. PF algorithm was proposed to approximately estimate the cost overrun probability in the hidden nodes of I/O HMM and to learn the unknown parameters and update the cost information in a real-time manner.

The conceptual diagram of the aforementioned model and its use is illustrated in Figure 1. Conventionally practical project control typically adopts earned value management (EVM) to assess the project cost and schedule performances [18]. In EVM, accumulated cost data are plotted as a trend curve, and the cost performance statuses are assessed based on the cost performance index (CPI). Once CPI reaches below 0.90, corrective measures need to be taken. Moreover, if future potential cost overrun events can be identified based on the existing cost trend curve, preventive actions can be proposed to reduce the project cost overrun risk. Classical EVM extrapolation approaches are used to explore future cost performance trends; nevertheless, consecutive cost statuses, influence factors with their independence, and the effect of corrective action are not included in the EVM extrapolation. To overcome the limitations of classical extrapolation methods, a DBN-based cost overrun forecasting model is proposed, and its usage is shown in Figure 1. The influence factors are modeled as nodes and their relationships are described using arcs at I/O HMM. The cost overrun to be predicted is modeled as hidden nodes and their state transition is assumed to follow Markov chain. The corrective action may be taken once project cost overrun occurs. The corrective actions are modeled as output nodes (also called as observation nodes in I/O HMM). Future cost overrun event probability estimated by the model provides warning information about the cost overrun risk. For the future cost overrun events with high possibility that are identified by the model, the proper effective cost–risk treatment action can be taken ahead of the occurrence of cost overrun.

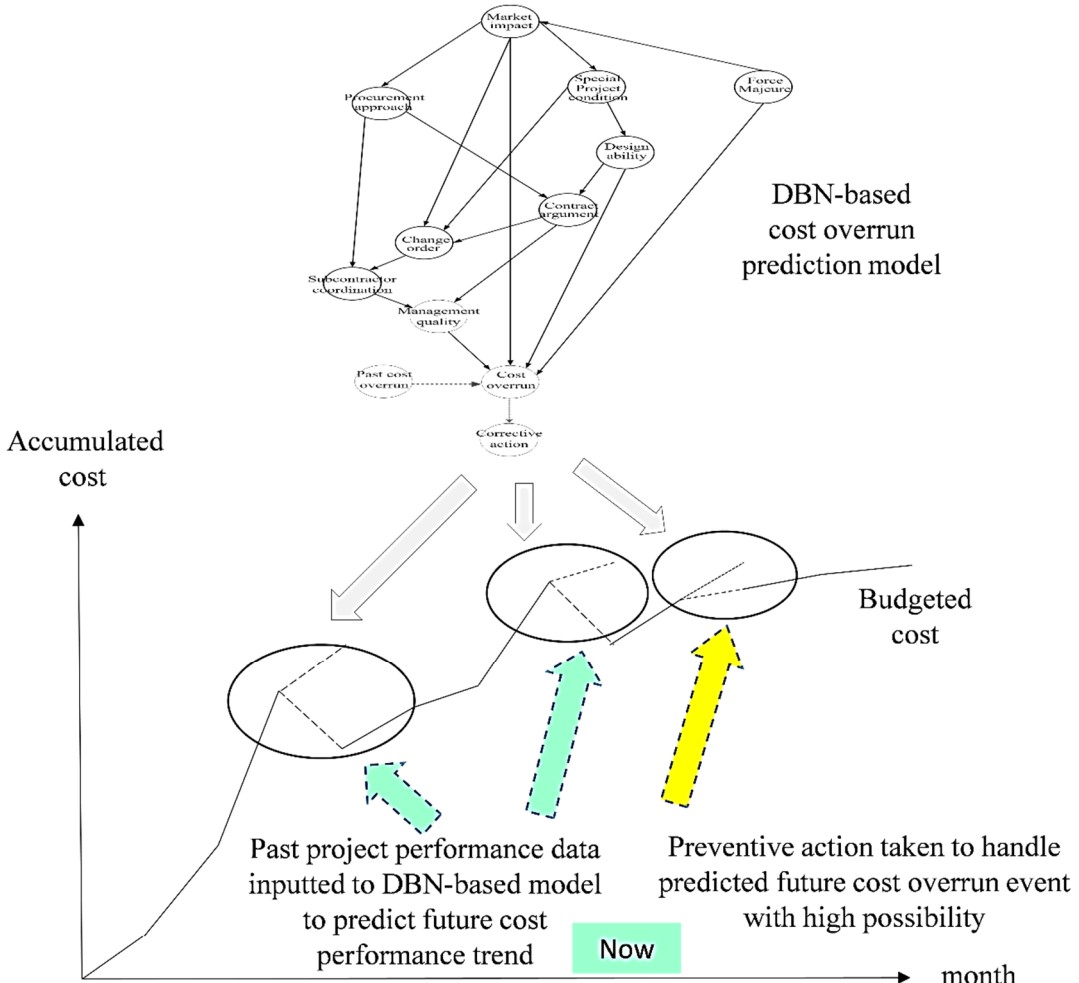

**Figure 1.** Utilization of proposed model during cost management process.

### 3.1. DBN and HMM

DBN is an extension of BN and it represents time influence of variables. These sequences are often based on time series or logical consequences at the problem domain. DBN consists of three fundamental theories: (a) probability theory, (b) graph theory, and (c) time series. Based on Bayes' theorem and full probability theorem, the static results (i.e., posterior probabilities at BN nodes) are first inferenced. The posterior probabilities at the different time periods are further estimated along with time sequence at DBN.

The hidden Markov method (HMM) can be regarded as the simplest DBN. For the purpose of effective analysis and inference, DBN is generally converted into HMM. There are four typical kinds of HMM suitable for DBN modeling (see Figure 2). Basic HMM consists of hidden nodes and observation nodes. Generally, the events to be predicted are modeled as hidden nodes and their consequence are described as Markov chain. Observation nodes are typically chosen to encode domain knowledge. The observation series is modeled based on the assumption that each observation only relies on an individual hidden state. HMM with mixture-of-Gaussians output (Mix-Gauss HMM) assumes the events follow Gaussian distribution. Auto-regressive HMM (AR HMM) is constructed under the condition that the observation nodes are time dependent. The last input–output HMM (I/O HMM) is used to model the problem in which some influence factors are formed as input to the hidden nodes and the sequence of observations is based upon the output states of hidden nodes. In our problem, nine influence factors were identified and organized as BN structure. The cost overrun events to be predicted are defined as the hidden nodes. The corrective actions may be taken once project cost overrun occurs. The corrective actions can be defined as the

observation (output) nodes in I/O HMM. In summary, I/O HMM is the appropriate model architecture to model our problem. Its functions are to predict project cost overrun risk and to assess the effect of corrective actions.

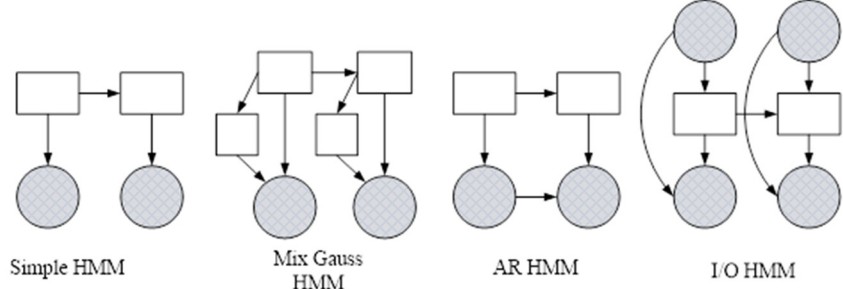

**Figure 2.** Four typical HMMs.

*3.2. Particle Filter (PF)*

PF is a sequential Monte Carlo simulation method, and it is a powerful tool for performing dynamic state estimation under noisy conditions using recursive Bayesian inference. It provides great, efficient, and flexible capability to approximate the systems with nonlinear functions and non-Gaussian noisy distributions [19,22–26]. Two factors that influence PF efficiency and accuracy are the particle number defined for the posterior distribution estimate and the transition function used to reassign these PF particles at the iteration.

As shown in Figure 3, in our study, PF was used to approximately explore the chance of cost overrun as the posterior probability based upon I/O HMM. To explore the posterior probability, the common PF procedure is as follows: using a collection of N weighted samples or particles, at time k, where $\pi_k^{(i)}$ is the weight of particle, a particle representation of this density:

$$p(x_k|\hat{y_{0:k}}) \approx \sum_i \pi_{k-1}^{(i)} \delta\left(x_k - x_{k-1}^{(i)}\right) \qquad (1)$$

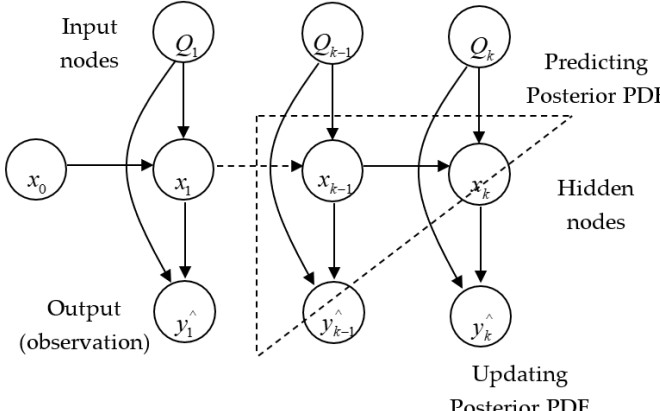

**Figure 3.** I/OHMM inference using PF.

Consider all the integrals needed to be performed at each filtering step:

$$p(x_k|\hat{y_{0:k}}) = \alpha P(\hat{y_k}|x_k) \int p((x_{k-1}|\hat{y_{0:k-1}}).p(x_k|x_{k-1})dx_{k-1} \qquad (2)$$

The recursive definition is used to compute the filtered distribution (1) given the distribution. With a particle representation for (2), it can be approximated as:

$$p(x_k|\hat{y_{0:k}}) = \alpha P(\hat{y_k}|x_k) \sum_i \pi_{k-1}^{(i)}.p(x_k|x_{k-1}^{(i)}) \qquad (3)$$

In this proposed model, importance sampling (IS) was adopted to create some appropriate sets of particles for representing the distribution of interest (3). It refers to a collection of Monte Carlo methods where a statistical expectation with respect to a target distribution is approximated by a weighted average of random draws from another distribution. Suppose that there is a density from which it is difficult to draw samples, but easy to evaluate for some particular samples, then an approximation to can be calculated as follows:

$$p(x) \approx \sum_i^N \pi^{(i)} \delta(x - x^{(i)}), \text{ Where } \pi^{(i)} = \frac{P(x)}{q(x^{(i)})} \tag{4}$$

Note that any distribution, known as a proposal distribution, can be used here, generally defined as uniform distribution. In overall PF, approximate inference of I/O HMM is composed of four steps as follows:

1. Draw N samples $x_k^{(j)}$ from the proposal distribution $q(x_k)$:

$$x_k^{(j)} \sim q(x_k) = \sum_i \pi_{k-1}^{(i)} p(x_k | x_{k-1}^{(i)}) \tag{5}$$

Uniformly select a random number $r$ over [0, 1], and then perform sampling from $p(x_k | x_{k-1}^{(i)})$ with the chosen particle $i$. This transition model assumes a linear Gaussian model, but any other model with easy sampling can be utilized.

2. Set the weight $\pi_k^{(i)}$ as the likelihood:

$$\pi_k^{(i)} = p(\hat{y_k} | x_k^{(j)}) \tag{6}$$

The samples $\left\{ x_k^{(j)} \right\}$ and above are evenly taken from $p(x_k | \hat{y}_{0:k-1})$. They are recursively reweighted in this fashion to account for evidence $\hat{y_k}$.

3. Normalize the weights $\left\{ x_k^{(j)} \right\}$

$$\pi_k^{(j)} = \frac{\pi_k^{(j)}}{\sum_m \pi_k^{(m)}} \tag{7}$$

4. Conduct I/O HMM inference:

Prior to the common procedure of PF, the transition probability of hidden nodes along with time series and the impact of the influence of BN factors on the hidden states should be integrated and normalized. Then PF, is used to approximate I/O HMM.

### 3.3. I/O HMM Model Procedure

The overall procedure of model construction from DBN to I/O HMM and PF and execution is depicted in Figure 4. The DBN's construction is composed of three steps: (1) the determination of BN nodes; (2) the elicitation of the dependence among BN nodes and their conditional probability tables (CPT); and (3) the conversion from BN to DBN. As depicted in Table 1, based on literature review and expert judgement, nine main factors were summarized and used to formulate BN. They are design ability, change order, management quality, subcontractor coordination, project condition, procurement approach, market impact, contract argument, and force majeure. They are formulated as basic BN nodes.

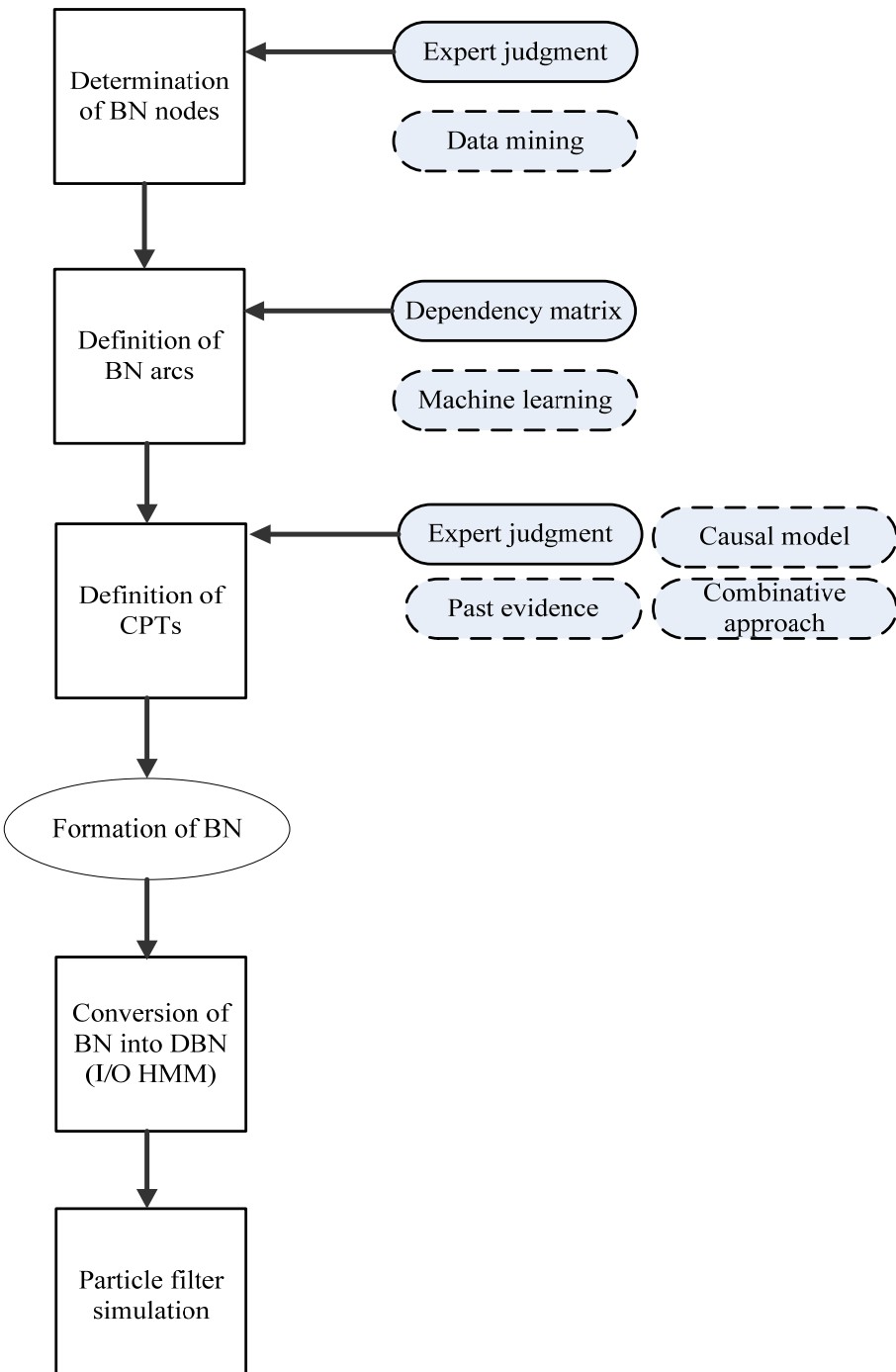

**Figure 4.** Overall procedure of model construction and execution from DBN to I/O HMM.

The second step of DBN construction is to define the relationships among BN nodes. This study applied the expert knowledge cause–result questionnaire by Hu et al. [27] to define the Bayesian network [17]. Mainly, the dependency matrix was utilized to collect the information about the node relationships. In the dependency matrix, the arrow stands for the dependencies from the parent nodes to the children nodes and the cross-out (X) mark means no dependency between nodes. This research interviewed four experts in the construction industry with over 15 years of practical experience. The dependencies among nine cost overrun influence factors are shown in Table 2 and the BN architecture was formulated based on Table 2 and depicted in Figure 5. The matrix is symmetric; the lower left cells are omitted.

**Table 2.** Dependencies among nine cost overrun influence factors.

| | Design Ability | Change Order | Management Quality | Subcontractor Coordination | Project Condition | Procurement Approach | Market Impact | Contract Argument | Force Majeure |
|---|---|---|---|---|---|---|---|---|---|
| Design ability | | × | × | × | ← | × | × | → | × |
| Change order | | | × | → | ← | × | ← | ← | × |
| Management quality | | | | ← | × | × | × | ← | × |
| Subcontractor coordination | | | | | × | ← | × | × | × |
| Project condition | | | | | | × | ← | × | × |
| Procurement approach | | | | | | | ← | → | × |
| Market impact | | | | | | | | × | ← |
| Contract argument | | | | | | | | | × |
| Force majeure | | | | | | | | | |

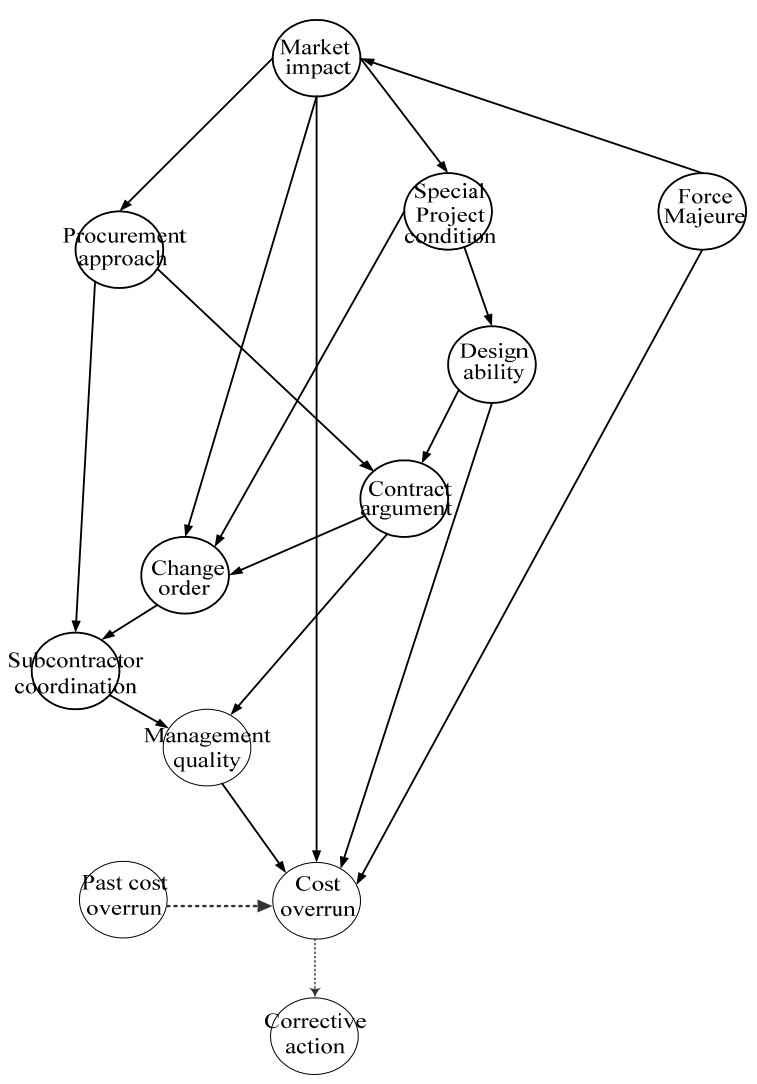

**Figure 5.** DBN architecture of cost overrun prediction model.

In addition to the BN structure, CPTs are required for BN inference. CPTs are generally elicited by past evidence, causal model, expert judgment, or combinative approaches [28–30]. This paper adopted expert judgment because of the constraint of past evidence. Nevertheless, CPT explosion could occur under the situation of several parent nodes and multiple states. For example, if a child node has four parent nodes and the number of their states is five, the total number of CPT values can be a great as $5^5$ (3125). In addition, the elicited probability values based on the experts' experiences could be conflicting, particularly under explosive CPT conditions. Several methods have been explored to overcome the large CPT problem [17,20,28]. This study adopted the ranked

nodes concept behind AgenaRisk to overcome the abovementioned difficulties [20]. At first, experts defined the weights among BN nodes. By coupling the weights with several predefined AgenaRisk parameters (typically AND and OR), the CPT values could be promptly estimated.

Basic DBN was formulated by converting static BN into dynamic BN based upon their temporal relationships. It is believed that the past cost overrun status will have an impact on the current status. Our research assumed that the temporal transition of cost overrun status follows a Markovian chain. Moreover, in addition to the cost overrun status sequence, the ongoing cost overrun status is also instantly influenced by the internal or external factors that were explored and formulated as static BN. Internal factors are the cost overrun factors that occur within the project scope, such as management quality and subcontractors' coordination. External factors are outside this project's scope, such as market impact and force majeure.

In the project control, the project managers needed to determine whether the corrective actions were required to lessen the gap between actual and planned performances after the occurrence of cost overrun. Practical project control typically adopts earned value management (EVM) to assess the project cost and schedule performances [18]. In practice, the warning level of cost performance index (CPI) is generally defined as 0.95. When CPI falls within range of 0.90 to 0.95, more intensive monitoring measurements are suggested. When CPI reaches below 0.90, corrective measures need to be taken for the prevention of the project cost going out of control. In DBN, corrective action was added into the model as the observation (output) nodes. Based upon the abovementioned discussion, the status of the corrective actions as the input to the observation node is defined into three levels: no action, intensive monitoring, and corrective action.

At the final stage, by combining the interdependent influence factors as static BN, cost overrun status as the hidden nodes, and corrective action as the observation nodes along with time sequence, I/O HMM architecture was formulated by transforming DBN, as shown in Figure 5. PF was then used for approximate inference.

## 4. Results

### 4.1. Case Description

Six real building and MRT construction projects in Taiwan with 53 monthly cost data in total were used for model validation and sensitivity analysis. The case background is depicted in Table 3. To more comprehensively survey the appropriateness of the model, data sampling was allocated to the different construction phases. Some are at the decoration stage, and some at the structure stage.

**Table 3.** Case Background.

| Project | Project Type | Construction Phase | Time to Be Surveyed | Project Description |
|---|---|---|---|---|
| 1 | School building | Decoration | November 2008 to May 2009 (7 months) | 11 floors and 2 basement floors Duration: 2007/4~2009/08 Floor area: 25,570 M$^2$ |
| 2 | Apartment building | Basement | February 2009 to July 2009 (6 months) | 20 floors and 3 floors in basement Duration: 2008/10~2011/07 Floor area: 45,660.6 M$^2$ |
| 3 | MRT construction | Decoration | October 2007 to May 2008 (8 months) | Duration: 2006/05~2009/03 Floor area: 18,915 M$^2$ |
| 4 | MRT construction | Decoration | November 2007 to June 2008 (8 months) | Duration: 2004/05~2009/02 Floor area: 24,854 M$^2$ |
| 5 | MRT construction | Structure | December 2008 to July 2009 (8 months) | Duration: 2008/11~2014/11 Floor area: 139,000 M$^2$ |
| 6 | Apartment building | Structure | March 2008 to June 2009 (16 months) | 18 floors and 2 basement floors Duration: 2007/8~2009/12 Floor area: 60,155.6 M$^2$ |

Rem: action is classified into three levels: no action/intensive monitoring/corrective action.

### 4.2. Model Validation

By conducting I/O HMM model inference with the input of real construction project data, the prediction results and their accuracies are shown in Table 4. The accuracy percentage is 86.8% overall. Three comparisons were performed for model validation. First, a project cost overrun risk prediction model using real-time Bayesian analysis was explored and constructed by Jenny (2008) [31]. Cost overrun event occurrence was assumed to follow a Poisson arrival pattern, and corrective action can be taken once poor cost status has occurred. The accuracy percentage of Jenny's model (2008) is 82.8%. I/O HMM model proposed in this study provides higher accuracy in cost overrun prediction.

**Table 4.** Prediction accuracy of cost overrun prediction model.

| Prediction Result | Number | Percentage (%) |
|---|---|---|
| Correct | 46 | 86.8 |
| Error | 7 | 13.2 |
| Type I error | 4 | 57.1 |
| Type II error | 3 | 42.9 |

Secondly, the cost overrun risk can also be estimated using static BN, in which past cost overrun status is neglected. Prior information was identified and input into BN to individually forecast the cost overrun status at each phase. The overall accuracy percentage of static BN is 82.9%. As shown in the following subsection, sensitivity analysis, past cost overrun status also plays an important role on the prediction. If such information is not covered in the prediction model, the model's accuracy may decline. In addition to model accuracy comparison, our model considers the combinative impact of the dependent influence factors that has rarely been explored in the previous cost overrun prediction models. Significant factors and their deferred effect can be identified based on the proposed model so that proper, effective cost–risk treatment plans can be more appropriately developed.

Finally, to compare project cost trend prediction with earned value management (EVM) on the same basis, EVM-predicted cost performances were converted to the cost overrun probability values and counted following a normal distribution, in which the overrun average and the standard deviation were estimated based upon project cost data. The percentage of accuracy of EVM with simple extrapolation is 76.8%, and our model was more accurate than EVM. In addition, our model also considers the effect of corrective action, which is hardly considered in EVM.

The project decision is also affected by the correctness of the model conclusion. As stated in Table 4, around 13% prediction error exists in the prediction model. A hypothesis test was further conducted to figure out the false positive and negative model conclusions (type I and II errors, respectively). There were seven predicted errors in total, in which four wrong predictions belong to type I error and three predictions type II error. Type I in particular error means that the prediction model indicates a low cost overrun probability; nevertheless, the real cost overrun does occur at this stage. In other words, with around 6% possibility, the model provides a more serious error message to project management because project managers may miss the chance to take timely cost overrun corrective action. Exploring the raw data in detail, it is found that the type I error generally occurs at after long and continuous cost underrun statuses. Further assessment and model calibration are recommended.

### 4.3. Sensitivity Analysis

One-parameter sensitivity analysis was further performed to study significant influence factors of cost overrun. Such information provides valuable support to an effective cost management program. Project managers can pay more attention to the significant factors based upon sensitivity analysis once cost overrun occurs. The sensitivity analysis result is summarized in Table 5. Management quality is the most significant cost overrun factor

overall. Subcontractor coordination is the second sensitive factor. The project manager can enhance management quality and set up an effective communication mechanism to coordinate subcontractors to prevent the occurrence of cost overrun under the condition of the cost overrun warning generated by the prediction model. Moreover, the past cost overrun status is also a significant indicator of cost overrun in the following project stages. If preceding cost overrun did occur, there is high possibility of the occurrence of cost overrun both at the present moment and in future. From the perspective of modeling, the information about past cost overrun status is important to the prediction's accuracy. It is better to build the cost overrun prediction model using a time-sequence approach, such as DBN.

**Table 5.** Sensitivity analysis results.

| Factors | Level of Sensitivity | Order |
|---|---|---|
| Management quality | 0.5282 | 1 |
| Subcontractor coordination | 0.2520 | 2 |
| Past cost overrun status | 0.2458 | 3 |
| Contract argument | 0.2257 | 4 |
| Change order | 0.2255 | 5 |
| Design ability | 0.2230 | 6 |
| Market impact | 0.1911 | 7 |
| Procurement approach | 0.1432 | 8 |
| Project condition | 0.1052 | 9 |
| Force majeure | 0.0645 | 10 |

## 5. Discussion

In theory, prediction models can generally be classified into two categories: the causal model and the time-sequential model. The causal prediction model must summarize the common significant causes to build the model. The time-sequential model mainly relies on the historical time series data of surveyed problems to build the model, such as EVM extrapolation and time series. Both have their own features and advantages. This study aimed to develop a prediction model by unifying the characteristics and the advantages of both the causal model and the time-sequential model while only employing the project's own cost data as input. This study used DBN to formulate problem architecture and to use I/O HMM with PF to run inference. The proposed model combined the time-sequential model, HMM, and the causal prediction model, static BN, to construct I/O HMM. Static BN, a causal prediction architecture, was used to assess the impact of influence factors on project cost overrun. HMM, a time-sequential prediction model, was used to simulate project cost performance trends based on project-owned cost performance status at each stage, and the corrective action once used. Overall, by unifying the characteristics and advantages of both the causal prediction model and the time-sequential prediction model, coupling with the consideration of the corrective actions, the model proposed in this paper showed better prediction results in accuracy, compared with EVM with simple extrapolation, HMM, and static BN.

However, some limitations in the model setting should not be neglected. Model calibration requires further research to improve the accuracy and usefulness of the model. First, model accuracy can improve if further adjustments are made for factors affecting cost overruns. Nine cost overrun factors and their interdependence were adopted to establish BN in this paper. As stated in the literature review, many internal and external factors could result in project cost overrun. A more comprehensive survey of influence factors and their interdependence can be carried out to build BN architecture.

Next, three BN construction approaches are generally used: (1) learning from a large amount of training data; (2) based on the experience of domain experts; and (3) a hybrid. The second approach is generally used for practical BN construction because of the constraint of data availability. Due to the constraint of research duration, four domain experts

were invited to explore BN node dependencies and CPTs. In the future, more domain experts should be invited to establish a more reliable BN.

Finally, the proposed model provided higher prediction accuracy compared with three different prediction models. The model would benefit by examining the more suitable cost overrun sequence model besides I/O HMM, as well as more realistic corrective actions and cost status assessments. As depicted in Table 4, there are still some type I and II errors. Further assessment and model calibration are recommended. In this paper, the corrective actions were defined as a status variable with three levels: no action, intensive monitoring, and corrective action. Practical correction action types and their effects could vary depending on the project scope and condition. Further tuning of these model parameters is recommended.

## 6. Conclusions

This study proposed a new method of project cost overrun risk prediction based on DBN. I/O HMM with particle filter was further used to run inference. The accuracy of the proposed model and algorithm was verified against six building and MRT projects in Taiwan. It provided higher accuracy in prediction compared with three other prediction approaches. Moreover, several significant influence factors of cost overrun were identified based on sensitivity analysis. They were management quality, subcontractor coordination, and past cost overrun statuses. This proposed method is able to provide fast and timely estimate of cost overrun probability based upon the preceding cost status and the interdependent influence factors of the DBN. This method supports a realistic preliminary prediction of project cost overrun risk. Based upon the output from the model, project managers can take proper effective cost–risk treatment action ahead of the occurrence of cost overrun.

**Author Contributions:** Conceptualization, S.-S.L.; Methodology, S.-S.L. and C.-Y.L.; Validation, C.-Y.L.; Formal analysis, C.-Y.L.; Writing—original draft, S.-S.L. and C.-Y.L.; Writing—review & editing, P.-L.W.; Supervision, S.-S.L. All authors have read and agreed to the published version of the manuscript.

**Funding:** This research received no external funding.

**Informed Consent Statement:** Not applicable.

**Data Availability Statement:** Due to the length of this article, the raw data for this study are not included in this paper but are available upon request. The data can be obtained by contacting the corresponding author at leuss@mail.ntust.edu.tw.

**Acknowledgments:** The authors thank the many experienced engineers who provided valuable information about construction cost management, Jianye Ching at NTU for modeling suggestions and the comments provided by anonymous reviewers.

**Conflicts of Interest:** The authors declare that they have no conflict of interest.

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
