# Peer review of "Dynamic-Bayesian-Network-Based Project Cost Overrun Prediction Model"

_sustainability, doi:10.3390/su15054570_

Round 1

Reviewer 1 Report

The authors chose an interesting direction as the object of research, namely the creation of a model for forecasting cost overrun.

The idea of using a dynamic Bayesian network is interesting and promising.

Unfortunately, the general level of the article does not meet the minimum requirements for publication.

The article turned out to be rather chaotic, the classic structure of the article cannot be maintained, the information between the sections is mixed up.

However, I see potential in the article and recommend that the authors fundamentally change the presentation of their research.

Specific recommendations.

1. Authors need to change the structure. I recommend using the classic approach, that is, dividing it into sections Introduction, Literature review, Materials and methods, Results, Discussion and Conclusions. All of these sections are mandatory in a revised article.

2. In the Introduction section, the authors need to interest the reader and lead him to understand the importance of this research.

Model overview is not needed in this section, move the information to Methods.

3. The literature review is poorly written, there is no analysis of research over the past few years, the influencing factors are selected from very "old" sources and are not described by the authors.

4. Authors should highlight the main findings of their work and explain how this study represents progress compared to other papers. Please clearly explain the novelty.

5. The modelling is well described. I recommend the authors to better describe the model building process, it is difficult to understand the transition BN → DBN → I/O HMM → PF.

6. There is also a question about experts and the mutual influence of factors. For example, the lack of correlation between Project condition and Management quality is surprising.

7. I don't understand why the authors added a paragraph about earned value management (EVM)? What does this have to do with this study?

8. Case study section. Described weakly and rather chaotically. The results appear without further explanation of how they were obtained. The important table 5 to which the authors refer is missing.

9. There is no Discussion section.

10. Designing the article.

The quality of some figures is poor, the inscriptions do not fit within the boundaries of geometric shapes.

Fig. 1. not clear. It is necessary to explain what the authors wanted to show with this figure

Fig. 2. is optional, it does not add additional value.

References to the literature.

Using such a style [5,18,21,28,29] is not the best option to explain the importance of the researched problem.

Aljohani et ah, (2017), Hu et al. (2007) ... please check the reference style.

Reviewer 2 Report

The authors need to explain the basis for the selection of 15 experts they interviewed. The criteria for classifying them as experts were not available. The sampling approach and adequacy of the size was also not explained. 

Though the candidate mentioned there were previous models for predicting cost overrun, they did not specifically evaluate the strength and weaknesses of at least the top ranked models in support of their assertion. 

Reviewer 3 Report

The paper need much more work and through investigation. The theoretical background is weak and literature review provides little recent sources of validation. Furthermore, the aim is not connected with the problem. The discussion is completely missing and the conclusions are really mediocre.

Here are the 

Essential Ingredients of a Publication

1.  Introduction:  presentation of the problem and clear explanation why this problem is important, and for whom it is important

2.  Research question: clear definition of the question and clear explanation why is this question important (who is waiting to obtain an answer to this question)

3.  Prior research: who has done work in similar areas, what were their successes and failures in trying to obtain answers to similar/related questions?

4.  Proposed methodology for your research: how will I answer my research question?

5.  Detailed description of research results, i.e. answer to your research question using the methodology as above

6.  Conclusions and directions for future research

Please rewrite and resubmit when ready.

Reviewer 4 Report

11 (9,27) Where does the 33% cost overrun value come from? What is the universe? The cited paper does not seem to mention that value.

2 (20-21,47-48,84-85,104-105,113-114,Fig1,236-237,305-307)

It is not clear what exactly is the result that is output from the model. Is it a simple Boolean overrun/no overrun prediction? Or is it a numerical estimate of the probability of cost overrun? Or is it a suggested corrective action? Or something else?

The relationship between the terms “corrective actions”, “output nodes” (observation nodes), and the actual output of the model is not clearly explained.

In Fig1 there are two items labeled “cost overrun”. They should be named distinctly and their relationship explained in the text body.

3 (49-51,54-55,58-59,84-85,94-96,104-105,112-113,140-144,Fig5,236-239,241,311-312)

What exactly are the corrective actions? Are they real actions to be taken by project managers? Or are they just the name of the observable (output) status variable?

If they are real actions, what are those, and can they be effective after a cost overrun was predicted and/or really occurred? If they are just a status variable, what do the statuses “no action”, “intensive monitoring”, and “corrective action” mean for project managers?

If corrective actions are only outputs of the model, how can their impact be assessed (for future model predictions)?

4 (48-51,140-144,229-231,235-236,293-296)

If the action is corrective, then the cost overrun (predicted or not) already happened, and the concept of overrun risk no longer makes sense. And if the model is trying to predict cost overruns (that did not yet happen), how can it take into account corrective measures since there is yet nothing to correct?

If the model is predicting future cost overruns based on a real cost overrun that already happened, it should be stated more clearly. Additionally, if the purpose of the corrective actions is to prevent a real cost overrun that already happened from getting even worse, how are these actions factored into new model predictions?

5 (96-97,269-270) When are the proactive and reactive cost risk management plans supposed to be generated?

6 (38-39,52,56,64,92-93,252-253)

Other than monthly project cost records , what is the exact nature of the data you use from project documents and checklists and how does it all relate to the 9 factors? Is it standardized and could it be collected from other construction projects (outside of Taiwan)?

How frequently can the model be applied/run (how often can the input data be easily collected)?

What are the estimated costs to collect the input data for each sampling point?

7 (155-187)

The equations are shown as poor quality images and should be replaced.

There following symbols lack definition/explanation: k (particle index), delta (discrete impulse).

Equation (8) is the same as Equation (7).

8 (205-206,Table2)

How did you conclude that four experts were sufficient?

The lower left diagonal should be filled with the “symmetric” of the upper right, or the text should mentioned that it was omitted.

9 (215-217) The concept of CPT explosion and its impact on eliciting probability responses is not explained.

10 (227-228) Which factors are inside and which are outside?

11 (244-298,Table3,Table4)

How many cost overrun situations occurred in each project? What was the distribution of cost overrun situations among each project? Were there obvious patterns such as long-winded sequences of overruns, or spurious single-overrun only situations?

What corrective actions, if any, were taken and when? How did they influence the model predictions?

What software tools were used for the simulation?

12 (289-293)

Where is Table 5?

How was the sensitivity analysis conducted?

The statement that project management can influence its own management quality and coordinate subcontractors is confusing. Furthermore, it may be in direct conflict with the implications of Figures 1 and 5: if the factors can affect each other bidirectionally (graph no longer acyclic) a BN is not an appropriate model. Can management quality really be perceived as a conditionally independent factor among the other 8?

What is the prediction accuracy if you run the model with only 2 factors: management quality and subcontractor coordination? Do your results justify the need for 9 factors?

13 (316) No data could be found in the paper to allow independent reproduction of results.

Round 2

Reviewer 1 Report

I recommend publishing the article in present form

Author Response

Your kind recommendation is deeply appreciated.

Reviewer 3 Report

The paper was much upgraded.  However, still the discussion is too weak and short. Please confront your results with the other similar studies you built your theoretical background on and explain what is your delta. Also explain the limitations...
